# Evaluating approaches to designing effective Co-Created hand-hygiene interventions for children in India, Sierra Leone and the UK

Sapphire Crosby[1]*, Sarah Younie[1], Iain Williamson[2], Katie Laird[3]

1 Institute for Research in Criminology, Community, Education and Social Justice, De Montfort University, Leicester, United Kingdom, 2 Institute for Psychological Science, De Montfort University, Leicester, United Kingdom, 3 Infectious Disease Research Group, Leicester Institute for Pharmaceutical Innovation, De Montfort University, Leicester, United Kingdom

* p13212007@my365.dmu.ac.uk

**Data Availability Statement:** All relevant data are within the manuscript. No Supporting Information files are required.

## Abstract

Effective and culturally appropriate hand-hygiene education is essential to promote health-related practices to control and prevent diseases such as Diarrhoea, Ebola and COVID-19. In this paper we outline and evaluate the Co-Creation processes underpinning a handwashing intervention for young children (A Germ's Journey) developed and delivered in India, Sierra Leone and the UK, and consider the implications surrounding Imperialist/Colonial discourse and the White Saviour Complex. The paper focuses both on the ways Co-Creation was conceptualised by our collaborators in all three countries and the catalysts and challenges encountered. Qualitative data have been drawn from in-depth interviews with five key stakeholders, focus group data from 37 teachers in Sierra Leone and responses to open-ended questionnaires completed by teachers in India (N = 66) and UK (N = 63). Data were analysed using thematic analysis and three themes, each with three constituent sub-themes are presented. In the theme 'Representations of and Unique Approaches to Co-Creation' we explore the ways in which Co-Creation was constructed in relation to teamwork, innovative practice and more continuous models of evaluation. In 'Advantages of Co-Creation' we consider issues around shared ownership, improved outcomes and more meaningful insights alongside the mitigation of risks and short-circuiting of problems. In 'Challenges of Co-Creation' we discuss issues around timing and organisation, attracting and working with appropriate partners and understanding the importance of local context with inherent social, economic and structural barriers, especially in low-and-middle-income countries. We consider how theoretical elements of Co-Creation can inform effective international public health interventions; crucial during a global pandemic in which handwashing is the most effective method to control the transmission of COVID-19. Finally we reflect on some of the methodological challenges of our own work and in managing the potentially conflicting goals of the ethical and participatory values of Co-Creation with pragmatic considerations about ensuring an effective final 'product'.

**Funding:** The authors received funding for this work from the Faculty of Health and Life Sciences, De Montfort University. The funders had no role in study design, data collection and analysis, decision to publish, or preparation of the manuscript.

**Competing interests:** The authors have no competing interests to report.

## Introduction

Hand-hygiene is a simple but highly effective behavioural strategy to control disease transmission and is amenable to intervention. This paper describes and evaluates a set of innovative educational resources and workshop activities ('A Germ's Journey') for young children initially developed in the UK which have also been adapted for use in Sierra Leone and India. To contextualise the study the introduction presents material around the importance of hand-hygiene to international public health. This is followed by a discussion of how public-health resources for young children can be developed through forms of community engagement and Co-Creation which we used at all stages of planning and refining the intervention. The prior research in which the intervention was developed and piloted is outlined briefly [1, 2] before a rationale for the current study's aims of adapting the resources to an international audience is presented. Challenges around the export of potentially post-colonial representations are reflected upon.

The importance of correct hand-hygiene has been well documented, with studies showing the percentage of deaths that could be avoided by performing appropriate handwashing measures. UNICEF state that diarrhoea is a leading killer of children, accounting for approximately 8 per cent of all deaths among children under age 5 worldwide in 2017. This translates to around 480,000 children a year. Almost 60 per cent of deaths due to diarrhoea worldwide are attributable to unsafe drinking water and poor hygiene and sanitation [3].

However, as UNICEF highlight, completing correct handwashing with soap 'can cut the risk of diarrhoea by at least 40 per cent and significantly lower the risk of respiratory infections' [3], highlighting the importance of good hand-hygiene and clean home environments.

Studies have highlighted the correlation between correct handwashing with soap and the reduction of acute respiratory infection [4–6] and the transmission of pneumonia and influenza among other infections [7–10]. Promoting hand-hygiene has been suggested to be one of the most effective methods of reducing infectious disease world-wide [11], yet it is estimated that just 19% of people globally practice sufficiently thorough handwashing after contact with excreta [12].

The ongoing coronavirus (COVID-19) pandemic has brought the importance of hand-hygiene in disease control and prevention to sustained worldwide attention. At the time of writing, non-pharmaceutical interventions, including undertaking correct handwashing practices are critical to disease control in the absence of an effective vaccine or antiviral medication [13].

With hand-hygiene being central to public-health and infection prevention, discourses around handwashing information and interventions have increased greatly. Information through both expert sources such as healthcare professionals and health officials, and lay sources such as mass and social media have the ability to either reduce the spread of infection by correct infection-control practises or misinform and/or detrimentally impact both the physical and mental health of the public [14, 15].

Ensuring that public health information is both accurate and targeted appropriately is vital, in order to correctly inform and aid understanding. Having noticed a lack of hand-hygiene resources targeted at young children in 2017, a set of innovative educational resources and workshop activities, 'A Germ's Journey' were developed for use in the UK. A team of researchers (including a microbiologist and educationalist) from De Montfort University, UK developed the resources with the aim to teach young children about germ transmission and handwashing. A wide range of child-friendly educational resources (including books, website games (www.germsjourney.com), posters, handwashing songs and colouring sheets) were specifically designed to be engaging and interactive, for children to use alongside their parents/

teachers. Information for adults about the topic and how to use the resources with their children/pupils have also been made available and shown to be an important element of promoting children's understanding and behaviours [1, 2].

An initial study [1] was conducted in the UK, in which Germ's Journey workshops were carried out with children in the Early Years Foundation Stage (EYFS) in order to evaluate the resources' effectiveness in aiding understanding. These workshops took the style of a carousel session, in which groups of children each took it in turns to complete four activities, each supervised by a trained adult (book reading, web-game activity, handwashing activity and colouring in). The interactive book reading activity involves children reading 'A Germ's Journey' [16] whilst web-games include a 'find the germs' game, in which children expose the 'hidden' germs within a cafe setting. An animated version of the book shows the consequences of washing or not washing hands after using the toilet. The handwashing activity involves children rubbing their hands with glow-in-the-dark gel as a visual representation of germs using ultraviolet light to see this. The children then wash their hands and use the UV light again to see if they have washed their hands correctly and if not, see the places on their hands that they have not washed [1]. Following positive results from this study in which learning resources were found to have increased young children's awareness and understanding of germs and handwashing since participating in Germ's Journey workshops [1] the researchers' next focus has been to develop culturally relevant educational resources for children in low-and-middle-income-countries, where knowledge of germs and correct handwashing practice can be life-saving [2]. Thus far, the book, posters and parent guides have been adapted and translated.

Having established the central importance of hand-hygiene in disease prevention and transmission reduction and outlined the genesis and initial evaluation of the resources in a British context, in this paper our attention now turns to discussing some of the well-established methodological frameworks used in developing effective public health interventions through community engagement, in India and Sierra Leone. The Germ's Journey project encompasses intersecting paradigms of Co-Creation, User-Led Design and Participatory Action Research [17]. For readers interested in our application of Participatory Action Research, please see one of our other publications [2].

## Co-Creation

Although first seen in academic literature twenty years earlier [18], substantive conceptual work is typically linked to Kambil and his co-authors who proposed 'Co-Creation' as a strategy to add value to businesses by working collaboratively with customers [19–21]. As Co-Creation has become more popular, it has arguably become more diffuse and the term is used in a variety of ways and contexts. Nonetheless, pivotal to contemporary understandings of co-creation is the principle that end-users play an active participant throughout the process. Thus Jansen and Pieters state that true Co-Creation is 'a transparent process of value creation in ongoing, productive collaboration with, and supported by all relevant parties, with end-users playing a central role' (p.15) [22].

Co-Creation has become a widely recognised approach to pedagogic practice as a means of actively involving individuals or groups of people in educational contexts including curriculum design or lesson planning. As a pedagogic approach, Co-Creation has links with various educational theorists and their critique on traditional educational approaches, including Dewey [23] and bell hooks [24]. In 'Democracy and Education', Dewey highlights the importance of the learner 'having a vital voice in any learning that takes place in the classroom and beyond' (p.1) [25] and being an active contributor in their own learning and development, as opposed to children being passive participants in the didactic approach to education at that

time. Similarly, bell hooks argued for a progressive and holistic approach to pedagogy, noting that a teacher's work is not merely to share information but to share in the intellectual and spiritual growth of [their] students. To teach in a manner that respects and cares for the souls of [their] students is essential if [they] are to provide the necessary conditions where learning can most deeply and intimately begin' (p.13) [24]. Furthermore, hooks emphasises the importance of an engaged pedagogy and mutual empowerment within the classroom, with educators and students sharing ideas and learning from one another. 'hooks defines engaged pedagogy as a reciprocal and vulnerable-making process for students and the professor, led by many voices, involving shared risk-taking and responsibility, and embracing the whole individual' (p.2) [26].

Co-Creation theorising has been flourishing, and a number of explanatory models of the concept that aim both to capture and inform Co-Creation processes have been advanced in recent years including the 'Co-Creation wheel' put forward by Ehlen and colleagues [27]. In this model the theorists identify 'urgency' as the starting point for Co-Creation activity and argue that equal amounts of attention need to be given to four processes for an effective innovation. These are labelled as 'construction' (the structure of the innovation), 'expertise' (the knowledge and other forms of capital available within the Co-Creation team). 'Relationships and emotion' (the affective elements of the enterprise include managing emotional aspects and relationships) and 'action' (all elements of the design and implementation of the innovation). The dynamic nature of the wheel implies that the process is continuous rather than linear with the varying components requiring consideration both in parallel and interaction, and on more than one occasion. However, whilst all the principles and processes are based on prior theory and research there remains a very limited database of application of models like this in the context of international health promotion and education. Effective co-creation involves maximising the different, often unique forms of capital that various collaborators can contribute [28].

In order to develop relevant and meaningful resources, both in terms of culture and age-appropriateness, it was of vital importance that the Germ's Journey team worked alongside a range of collaborators, each with different skill sets and expertise. By merging the key ideas behind Co-Creation in both education and business/marketing contexts, the Germ's Journey team developed a number of different educational resources for use in diverse contexts and environments. The Co-Creation process with each collaborator will be discussed in greater detail in the methodology section.

In terms of this research project, to develop and maintain relationships with these collaborators to co-create relevant resources for children and to foster the embedding of the resources within each community, enabling the sustainable use of the resources long-term once the researchers have completed the study and were are no longer in-situ. Thus, each educational resource and workshop was created with both children and teachers/parents in mind as the 'end-users', therefore, it was important to work alongside children and professionals who are experts in children's learning development.

Research incorporating Co-Creation has primarily, at least in a health education context, been based in the most developed countries. Inherent within using these approaches in a genuinely international context are various socio-historical and political tensions around power relations especially where the research team is primarily from privileged White cultural backgrounds. As part of the Germ's Journey project team-members visited areas of socio-economic disadvantage in India and Sierra Leone in order to Co-Create cultural relevant educational resources. Therefore it is important to be aware of the narrative surrounding Imperialist/Colonial discourse and the White Saviour Complex.

Aronson notes that 'in 2012, Nigerian-American novelist Teju Cole coined the term white saviour industrial complex (WSIC) [29]. Anderson describes WSIC as being the 'confluence of practices, processes, and institutions that reify historical inequities to ultimately validate white privilege' (p. 39) [30]. Aronson highlights that popular discourse surrounding the Western World's view of Africa, for example, as a 'chaos, war-thirsty people, and impoverished HIV-infected communities, situates these countries as places in need of heroism' (p.36) [29], creating the mindset that certain areas and other low-and-middle-income countries are in need of external forces to come to the aid of communities in these areas, without regard for the role that colonialism and white supremacy have had in creating these situations. Allen [31] notes that inaccurate narratives that portray countries in Africa or low-and-middle income countries allow for 'the hegemonic project of whiteness and white supremacy' (p. 36) [29] to furnish a need for 'white intervention', leaving the agency of the individuals or communities in such settings to be disregarded, or non-existent [32]. This leaves the agency with the 'white saviours' and not the countries citizens to be empowered to act in their own interests.

In the case of this study, it was important to be aware of this narrative and sensitive to the implications of how a team of researchers from the UK visiting low-and-middle-income countries to develop resources for a handwashing intervention may be conceived. The researchers were aware and had to make clear that they were not visiting to 'teach children and teachers how to wash hands properly' but instead in keeping with the underpinning principles of the project, worked in collaborative partnership with the researchers in developing the resources ensuring that their shared motivation, knowledge of culture, and context informed all aspects of intervention design, adaptation and evaluation. The whole ethos of the Germ's Journey project is to collaborate with the individuals and communities in whom the resources will be used, as opposed to creating and donating resources, based on our Western ideologies and viewpoints. It was not a case of visiting areas such as India and Sierra Leone and teaching children and teachers 'the way that we do it in the UK'. On the contrary, it was a collaborative process which involved many conversations with many individuals and extensive research of the areas to ensure that the resources would be authentic and culturally relevant.

Therefore, building on the findings from the team's previous papers that narrate the initial development of Germ's Journey and a quantitative evaluation of the Indian arm of the intervention [1, 2]. In the current article we focus on the conceptual and practical considerations of Co-Creation in an international context. To do this we present and interrogate qualitative data from a variety of sources to explore how Co-Creation was viewed and evaluated in Sierra Leone and Gujarat, India (culturally very different contexts with a shared public health agenda around child mortality around diarrhoeal diseases) as well as the United Kingdom.

### Research questions

What were the experiences of Co-Creation partners on a children's hand-hygiene intervention in Sierra Leone, India and the United Kingdom?

More specifically, how was Co-Creation constructed by partners and what were the reported strengths and weaknesses of the Co-Creation process.

### Methods and materials

Qualitative data were collected from a variety of sources and analysed using thematic analysis. The core analysis is based on interviews with five key stakeholders who were involved in development of resources for the UK and adaptation of resources to the Sierra Leonean and Gujarat contexts. These core data are supplemented with focus group data (N = 37) collected from five

groups held with early years educators in Makeni, Sierra Leone and data from open-ended questionnaires collected from early years teachers in India (N = 66) and the UK (N = 63).

## Data collection

**Semi-structured interviews.** Seidman highlights that the purpose of interviewing is to listen to and understand "the lived experience of other people and the meaning that they make of that experience" (p.9) [33]. For this study, semi-structured interviews were conducted with five collaborators who, at different stages, have been involved with the development of the Germ's Journey educational resources.

The use of open-ended interview questions mitigates against investigator bias and enabled the researcher to receive more in-depth information [34]. This approach facilitates a more conversational discussion, which equates with the philosophical approach and methodology of having research 'done with' participants, rather than 'done to' them [35].

The questions focussed on each collaborator's personal approach to using Co-Creation within their workplace, including experiences, and opinions of the strengths and limitations of the method, both generally and in relation to the Germ's Journey project.

The interview questions structured discussions between the researcher and collaborators and were provided to the participants ahead of the interviews which made the purpose of the interview wholly transparent and allowed them the opportunity to prepare for the interview effectively. The schedule was used flexibly during the interview to follow the flow of the conversation. Interviews were completed either on the phone or online using VOIP software and lasted approximately 30 minutes. Interviews were conducted by SC and recorded to allow for transcription. Interviews were agreed to be completed in English as all participants were fluent in the language.

**Focus groups (Sierra Leone).** Using open-ended, semi-structured questions, five focus groups of seven or eight primary school teachers per group (total N = 37) were conducted at the University of Makeni, Sierra Leone. These groups were held after participants had experienced a workshop using the UK Germ's Journey resources with the teachers and primary school children and were planning the West African version of the Germ's Journey book. Questions focussed on what content and illustrations should be included in this version of the book as well as discussions around practical considerations around handwashing facilities and practices in local schools.

**Questionnaires (India and UK).** Open-ended questionnaires were completed by teachers in the Gujarat State of India (N = 66) after participating in trainer workshops using the UK version of the resources, in order to explore how to develop culturally relevant Gujarati resources.

Similarly, questionnaires were completed by primary school teachers (N = 63) in the UK in order to evaluate the Germ's Journey book, website and other workshop activities.

**Collaborators.** Interviews were conducted with five of the key collaborators involved in the process and development of Germ's Journey Educational Resources. Table 1 presents each

**Table 1. Collaborators.**

|       | Collaborator | Location | Co-Created Resource |
|-------|-------------|----------|--------------------|
| **Alex** | Learning and Engagement Officer | Thinktank Birmingham Science Museum, UK | Handwashing song/video, |
| **Ben** | Assistant Director: Office of International Relations and Projects | University of Makeni, Sierra Leone | West Africa book |
| **Chahel** | Project Co-ordinator | Manav Sadnha, Ahmedabad, India | Gujarati poster and book. |
| **Dan** | Marketing | PAL International, UK | Soaper Heroes. |
| **Emma** | EYFS Teacher | Primary School, Leicester, UK | Children's worksheet |

collaborator, their location and the learning resource in which they co-created alongside the Germ's Journey Team. Each of these have been suitably anonymised with a pseudonym for ethical reasons.

A summary of each collaborator and how they became involved as a collaborator is provided below to contextualise their accounts.

Alex became involved in the Germ's Journey project, when as part of a recent refurbishment, Thinktank Birmingham Science Museum designed a new interactive STEM (Science, Technology, Engineering and Mathematics) gallery named Mini Brum for children under eight-years-old. The gallery was Co-Created alongside schools, families and community groups with children playing a significant role in the process. The Germ's Journey team worked collaboratively with Thinktank to develop an educational handwashing song (https://www.youtube.com/watch?v=kLYcRFvyH3E&t=3s) that presents a step-by-step guide on how to wash hands, emphasising the areas of the hands that are often missed when handwashing. The song was also co-created by musicians and children at a local primary school, in which the children were involved in writing the lyrics to the song, in order to ensure the phrasing was easily understandable for children. The song was filmed and is now played in the toilets on a 'magic mirror' (mirror with a button that once pressed, plays the video), which were specifically designed to have the song/film as an interactive feature to encourage correct handwashing. The museum also run Germ's Journey educational workshops for school trips.

Ben is the Assistant Director for the Office of International Relations and Projects at University of Makeni in Sierra Leone. Ben worked alongside the Germ's Journey team to develop a West African version of the Germ's Journey book, so it was suitable for use in Sierra Leone and other areas of West Africa. The University of Makeni and local teachers were involved in the process of the book's development, including the images used in the book to authentically represent the types of food, toilet and handwashing facilities available in the area.

Chahel is a project organiser at the charity Manav Sadhna in Ahmedabad, India. Manav Sadhna is a charity 'engaged in constructive humanitarian projects that cut across barriers of class and religion while addressing issues faced by socio-economically neglected segments of society' (Manav Sadhna, n.d). The Germ's Journey team first established an ongoing working relationship during a visit to Ahmedabad in 2017, where the Germ's Journey team worked alongside Manav Sadnha to deliver handwashing workshops with local children and training workshops with teachers. Since this visit the two teams have worked collaboratively together to develop culturally relevant posters, songs and a Gujarati version of the original 'A Germ's Journey' book.

Dan works at an International Cleaning Products Supplier based in Leicestershire and became involved in the project in order to Co-Create educational resources and handwashing products that have been specifically designed to be used in paediatric wards in an upcoming research project to commence later on in the year. The research project aims to study the effectiveness of the educational intervention 'A Germ's Journey Soaper Heroes' which are hand-hygiene resources centred on superhero type characters. These are being implemented on paediatric wards, and evaluation of pre and post handwashing practice and understanding of germ transfer will be conducted.

Emma is an EYFS teacher at a primary school in Leicester. Emma worked alongside the Germ's Journey team in order to Co-Create an assessment tool to be used in a research study in which educational workshops were delivered to children in Ahmedabad, India (Crosby, Laird and Younie, 2019b). The tool took the style of 2 identical worksheets in order to obtain a baseline assessment and post-workshop assessment of the children's level of knowledge, to evaluate whether the intervention had an impact on improving children's knowledge of pathogen transmission and handwashing.

**Data analysis: Thematic analysis.**   In order to analyse the transcribed data obtained in the interviews, focus groups and questionnaires, a thematic analysis method was implemented following Braun and Clarke's six steps of undertaking thematic analysis: Namely, 'Familiarizing yourself with your data', 'Generating initial codes', 'Searching for themes', 'Reviewing themes', 'Defining and naming themes', and 'Producing the Report' [36]. For this study the researchers followed the six stages in order to correctly complete a thematic analysis of the written data obtained in the interviews, focus groups and questionnaires. Initial themes were developed and tabulated by SC, subsequently refined by IW and audited by all four authors.

**Ethics.**   The study received ethical approval from De Montfort University's Health and Life Sciences Research Ethics Committee. The study adhered to the British Education Research Association ethical code of practice [37]. Informed consent was obtained (both written and verbal) by the project's collaborators. All researchers reside in the UK and thus no formal permits were necessary. De Montfort University has a formal working relationship with the international collaborators in this study, and a Memorandum of Understanding (MoU) has been completed between De Montfort University and University of Makeni, Sierra Leone and a public engagement and research contract is in place between De Montfort University and Manav Sadhna, India acting as an equivalent of such permits.

## Findings

Following the interviews with the collaborators/participants, certain themes have been identified and have been presented in subsequent sections (Table 2). All themes will be represented by anonymised quotations primarily taken from the interviews and supplemented by some quotations from the focus groups and questionnaire data. Quotations are from interviews unless indicated otherwise. The themes are described in this section and then considered more broadly in the discussion section which follows.

### Representations of and unique approaches to Co-Creation

When asked how each collaborator defined Co-Creation, varying responses were given. The following sections have been named using the specific terminology used. The Collaborators also gave an insight into the distinctive ways in which they use Co-Creation in their practices. These are summarised in the three subthemes which follow:

**Co-Creation as *'more than teamwork'*.**   Whilst teamwork analogies were commonly used in the data, two elements were reported to enhance that element to make the enterprise more productive—ensuring that all participants and end-users were active throughout the process and through identifying more systematic ways of working:

> *"For us it's following people's interests, developing partners and working together to achieve the best possible outcome."*

(Alex, UK)

**Table 2. Findings and discussion themes.**

| Representations of and Unique Approaches to Co-Creation | Advantages of Co-Creation | Challenges of Co-Creation |
| --- | --- | --- |
| Co-Creation as 'more than teamwork' | 'Shared Ownership' and Inclusion | "Timing and Organisation" |
| Co-Creation as 'collaborative innovation' 'with multiple stakeholders' | "Better Outcomes" and 'More Meaningful Insights | Ensuring the "Right Contributors" |
| Co-Creation as a 'different approach to evaluation' | Short-Circuit Mistakes" / "Risk Mitigation | Understanding the Local Context |

Alex's organisation had considerable experience in using Co-Creation centring on the views and input from their end-users (in this case children):

Following people's interests' was an integral part of their Co-Creation process, with Alex noting that it was important that: *"What we are co-creating appeals to the people that we are working with because if they're not interested, it's not going to be a fun or fruitful process because they're not going to want to be involved."*

Similarly to Alex's definition of Co-Creation as 'working together', Dan also defined Co-Creation as 'teamwork' and 'sharing' but, perhaps given the business background, were focused on capturing the protocol and the end product alongside the process.

*"In some ways the label 'Co-Creation is just a different way of, what for years in a business like this we'd be calling 'team work'. So it isn't necessarily a new way of working but within teamwork what tends to happen is tasks are broken down into their constituent parts, and people go away and work on the individual parts independently and then come back together with the whole. What I understand Co-Creation to be is everybody being involved in the different parts. So, we're talking about a higher level of co-operation and teamwork, than historically we've had. "It means 'sharing', whatever that deliverable may be and working on it as part of a team where everybody has input and involvement into all stages"*

(Dan, UK)

Dan explained that his company had used Co-Creation in order to develop products for their customers but previously more informally.

*"Our philosophy historically has been 'teamwork is good', so it's been encouraged, but we don't have a formal process for it, [or a] project management approach that drives co-creation."*

(Dan, UK)

However, Dan's views on effective Co-Creation had evolved somewhat into advocating a more organised (and arguably less organic) process and method of using Co-Creation in more reliable and systematic ways in order to *"make it more of a repeatable process"*.

**Co-Creation as *'collaborative innovation with multiple stakeholders'.*** Data from all countries showed an appreciation of the creativity and innovation of Co-Creation.

*"To us Co-Creation is a collaborative development of new value (concepts, solutions and services) together with experts and/or stakeholders. It is a form of collaborative innovation in which ideas are shared and improved together, rather than kept to oneself".*

(Ben, Sierra Leone).

Emma gives a similar description, defining Co-Creation as *'working collaboratively'* emphasising the importance of utilising her colleagues' different skill sets:

*"Ultimately it's working collaboratively to develop resources or experiences for the children. We generally look at it as adults working together. [Co-Creation is] "coming up with an idea or something you want to achieve and it's using different peoples' skills and ideas to contribute towards and create whatever your goal is, and then a series of feedback and decision-making to come up with a final product or idea or experience for the children".*

(Emma, UK)

Likewise, when asked what their definition of Co-Creation was, Chahel commented on Co-Creation's ability to combine the innovative ideas of various individuals contributing to the process:

*"Co-Creation is a very useful process which effectively helps organisation. We can combine innovative ideas of different people and bring it on one platform to the betterment of the community and society."*

(Chahel, India)

Contrasting with Emma's approach, in which teachers collaborate to plan lessons for the pupils, Ben works with undergraduate students, noting that the main aim of using Co-Creation within their practice is to enable students' voices to be heard in the process of developing a curriculum. This was further enhanced by students being encouraged to develop and contribute their own resources to supplement those of the University itself:

*"The central aim is to create a curriculum that brings innovation and creativity. Students and teachers are both motivated to help each other create an experience that enhances study. Students' resources are integrated with organizational resources to facilitate a range of activities and experiences that encourage exchange and interaction which can lead to better practice and innovation."*

(Ben, Sierra Leone)

Whilst Ben looked at older student involvement, like Alex, Emma favoured the idea of working with pupils to Co-Create lessons:

*"I think at school we don't use all the different potentials, we don't use children in our planning and I think, you know, it's for them so why don't we get them more involved in it? We give [the children] information but we don't create with them. Maybe there's more of an opportunity. [The interview questions] made me look a little bit more about what we're doing and how we could change and be more inclusive when we are creating things."*

(Emma, UK)

Chahel uses Co-Creation predominantly as part of developing educational outreach programmes and activities for children and other members of the community with various stakeholders and contributors. When asked for specific of examples of when the organisation uses Co-Creation, Chahel explained:

*"Through the organization, particularly teachers, coordinators and experts of education jointly for education, art and craft or dance workshops etc, we use the Co-Creation."*

(Chahel, India)

The inclusion of multiple stakeholders was shown amongst the teachers in Sierra Leone who attended the workshops following initial implementation were similarly enthusiastic when the workshops sessions were piloted in class with children of three to seven years.

*"The workshop was creative and active, everyone was involved"*

(Teacher X, Focus Group, Sierra Leone)

**Co-Creation as a *'different approach to evaluation'*.**    Whilst discussing how her team uses Co-Creation within their practice, Alex cited a colleague

*"My colleague said something that was really good as well, she talks about Co-Creation being a sort of different approach to evaluation, and evaluating at the start rather than just sort of creating something and evaluating at the end. It's asking people to begin with and them being involved in that journey with you, getting their ideas from the start and then working on those rather than, you know, doing something and then asking people to comment after".*

Although all collaborators gave a similar definition, as expected the way in which they each use Co-Creation within their individual practices does differ. Alex's agency uses a multi-faceted Co-Creation approach, collaborating with a variety of individuals and groups whilst still keeping a child-centred approach/ Dan highlighted the importance of transparency when working as a team, and sharing ideas with people of different skills sets when working on a project. Likewise, Emma works alongside a team of fellow teaching staff to develop lesson plans, however, as a result of working with the Germ's Journey team, is keen to involve pupils in this process more. Chahel uses Co-Creation to develop activities for the local community and Ben works with the university students in order to co-create a beneficial student environment and curriculum.

## The advantages of Co-Creation

**'*Better outcomes'* and *'more meaningful insights'*.**    When discussing the advantages of using Co-Creation within the company, Dan noted that:

"*Ultimately, we get more out of the resources that we have available to us so either a higher volume of work or ultimately a better product at the end of it and better outcomes at the end of a process.*"

(Dan, UK)

With regards to the cleaning supplier company's involvement with the Germ's Journey project, Dan stated that:

"*I think informally we've created quite a good level of communication and involvement of lots of different people and because of that we've ended up creating something which I think is of greater value to everyone involved. So for us, rather than just being a corporate social responsibility programme it's actually now an important part of our marketing and content strategy, it informs part of our product development strategy. It's been really good.*"

Speaking specifically about Thinktank Birmingham Science Museum's involvement with the Germ's Journey project, Alex noted that the process was:

"*Absolutely fantastic it's been a brilliant partnership, I think it's one of the true successes of Co- Creation is that it's not just creating the products and then leaving it, it's ongoing . . . the song [we created] with the children, which is brilliant, it's in the gallery now and it's really successful. You can tell by people's reactions that it is pitched right and I think that's down to the Co-Creation process and sharing the skill sets. And we very much incorporate some of the things that you did into our sessions [educational workshops with children]*"

In addition to outreach programmes and activities, Chahel noted that the charity organisation have used Co-Creation to develop a book for children, commenting on the beneficial impact that this approach has had on the children.

*"Previously, we have designed a value education book using Co-Creation. Through this book we try to inculcate small values amongst the children using various mediums. These values are: respect elders and parents, love animals and nature, about sanitation, cleanliness, love, compassion etc. We are using these tools very effectively as a part of education, and children enjoy learning this and at the same time they carry the message as well".*

Chahel also commented on the direct outcome of the book that was co-created with the Germ's Journey Team, Manav Sadhna (charity organisation) and children and teachers in the local community in India with an apparent improvement in the rates of hand-hygiene:

*"The Germ's Journey book is a beautiful example of Co-Creation. We used this book with our Anganwadi kids (pre-school) of 3 to 5 years and also with kids of 6 to 10 years. Often our teachers use this book to spread awareness about the importance of handwashing in daily life, and they are successful passing this message amongst the children and through children the message will reach to their parents. We noticed that the majority of children have now adopted the habits of hand washing not only at the centres but at home as well. I think the habits adopted in childhood will remain forever."*

(Chahel, India)

Likewise, Emma highlights that one of the advantages of using Co-Creation to develop lesson plans, is the positive outcome it has for the children.

*"Topic planning is one of the biggest things that we do through Co-Creation, and it's nice to see the children having different experiences. It makes it like a richer experience for the children". We've got lots of different skills and approaches and it gives the children a more authentic and rounded experience we feel, more so than one person who doesn't have the experience coming up with ideas".*

(Emma, UK)

Ben notes that:

*"Co-Creation in my experience gives more meaningful insights; it has ears to the ground. Insights obtained from learning from each other directly can bring so much value to education."*

(Ben, Sierra Leone)

Furthermore, Ben highlights the importance of using Co-Creation within his university, in order to establish a collaborative, working partnership between the students and staff at the institution.

*"The characteristics or criteria for successful Co-Creation includes the respect for students, importance of students' active participation and their openness to contribute and create value in the educational process. The process of* Co-Creation *can allow institutions and*

*students to work together to improve student experience and enhance students' ability to act as partners'".*

(Ben, Sierra Leone)

Teachers in all countries also spoke about the benefits they perceived to their pupils of the co-created resources, especially the book and website:

"*The book together with the pictures provides more understanding and helpful in their lives*"

(Teacher X, India, questionnaire)

'*cleverly done, colour changing paint instantly interests the child, bright bold pictures. This then draws them into wanting to read. Think would be an especially good book for pre-schoolers / foundation year. Website engaging, easy to use*

(Teacher Y, United Kingdom, questionnaire)

**"Short-circuit mistakes"/"Risk mitigation.**   Dan highlighted the importance of transparency and visibility in the Co-Creation process and how consequently this decreases the chance of mistakes.

"*We now have much more of a Co-Creation approach which is 'everybody has sight of all stages of the process and can input directly without waiting for that sequential process to reach them. So, to facilitate that we use Microsoft Teams, and we use Trello boards, so that people can see it at any point, they can dip in and out if they want to. But what that means is we short-circuit the mistakes that we would have otherwise made*"

Similarly, Ben comments that "*Risk Mitigation is another advantage in Co-Creation*". By working alongside collaborators with different skill sets and continually evaluating the process and 'end-product', the element of 'risk' is subtracted because mistakes are more likely to be identified earlier on.

## The challenges of using Co-Creation

Various challenges to effective Co-Creation were also observed and these are considered in the final theme. These were around timing and organisation, bringing in appropriate and committed local contributors and understanding some of the challenges (socio-political and economic) of the milieu in which Co-Creation was occurring.

**"Timing and organisation".**   Alex highlights the importance of organising time efficiently:

"*It's definitely worthwhile that's what I would emphasise but it does require a lot of work depending on the groups you're working with for example, working with schools, and time is obviously very precious in school*".

(Alex, UK)

Similarly, Emma notes the amount of time that goes into the process can be challenging. Speaking of using the process when creating lesson plans:

*"It can be more time-consuming, so if you have a particular task and do it on your own it can sometimes be a lot quicker than trying to get people's ideas and then justification for what you're putting in there or not putting in there".*

(Emma, UK)

Whilst collaborators were committed to the value aspects of Co-Creation, good outcomes were essential for the significantly increased resources that were invested (and typically diverted from other activities, in Co-Created processes.

**Ensuring the "Right contributors".**   Two of the collaborators in their interviews emphasised the need to ensure that the individuals or groups that are being worked with can add value to the process, in that, the skill sets of these people are utilised where they are best-suited:

*"Ensuring you're working with the right group that's definitely what we've learnt, like for us working with young children and defining what they do, so just being careful you're asking the right people to help with the right things".*

(Alex, UK)

*"The risk with [involving lots of people] is, you have a lot of voices and you can potentially get conflicting feedback, almost a sort of paralysis through too much information. So I think it's just a case of having to manage that which really for us has been making sure that the right people are involved in the project rather than necessarily going 'everyone is involved in this'. So I think for us the risk would be if you involve too many people, get too many opinions that end up taking too long to sift through it. I think even with the Co-Creation process and the tools that we have to streamline that process, you still have to make sure that you've got the right contributors because if we involved everybody in everything we'd never get anything done"*

(Dan, UK)

Dan notes the need to strategize and balance the democratic and relational aspects of the process of Co-Creation with ensuring that some degree of restriction on both contributors and contributions to ensure that ultimately the enterprise remains productive.

For Emma the principles of inclusion, whilst worthy, had to be balanced with strong leadership. However, she recognised the challenges of matching the twin goals of maintain the ethic of 'Co-Creation' with production of an effective product:

*"The sorts of problems that we would have, is mainly down to communication and not having a clear form of leadership sometimes within it. I know it's supposed to be, well my idea of creation is, contribution. Lots of people contribute, but personally I do feel it needs some leadership as well and that's the balance between that, that's the difficulty."*

(Emma, UK)

For Ben, a rather different threat was apparent:

*"One key challenge is the commitment of partners and stakeholders- Getting Ministry of Education officials and Municipal authorities to do the needful that will promote Co-Creation is a teething challenge".*

(Ben, Sierra Leone)

Although more likely to get a better end-result by using the process, the on-going, collaborative nature of Co-Creation means that it is a time-consuming process. A great deal of attention and organisation is required to ensure true Co-Creation and consequently result in a meaningful and successful deliverable. The balance of who and how many people to involve, so as not to stunt the progression yet still ensuring an inclusive process, alongside ensuring the process is driven forward by somebody taking the lead role can, as noted by the collaborators, pose a challenge, as can the selection of less motivated partners.

**Understanding the local context.**   A final challenge in ensuring effective Co-Creation and positive outcomes was ensuring that all activities and resources were cognisant of and sustainable within the social, structural and economic challenges of the local context, especially if that was particularly impoverished. This element was raised most significantly in the Sierra Leonean context and was influenced by the ongoing limitation in availability of key resources in schools which was identified following the Ebola crisis in the country in 2014 and 2015. Several schools reported a lack of hand-hygiene resources including sanitiser, soap and 'Veronica buckets', commonly used in parts of West Africa without plumbed running water to ensure hand-washing in undertaken in flowing not stagnant water. Several of the teachers in the focus groups, whilst being universally positive about the educational workshop itself commented on this:

"*but the only thing the resources like soap, buckets are limited*"

(Teacher K, Focus Group)

"*the workshop was good but the school needs support for the resources*"

(Teacher Q, Focus Group)

Similarly, in questionnaires completed by teachers in India, among the positive comments about the workshop and resources, requests were made for resources ranging from microscopes to soap to be provided:

"*Please give us machines in the playgroups which would help us see the germs. Also give us hand wash*"

(Teacher J, India, Questionnaire) . . .

These concerns do not negate the very positive effects of Co-Creation but need identifying at an early stage of resource development. Co-creation activities need to consider some of these fundamental aspects early in inception and development.

## Discussion

In this paper we have investigated the understandings, facilitators and challenges involved in Co-Creation in an international context.

### Representations of and unique approaches to Co-Creation

Co-Creation as an approach is distinct from other methodologies due to its ability to include and give a voice to all participants, with their contributions being valued equally throughout the process. With regards to this project, this is important for a number of reasons. Firstly, when developing resources, particularly for low-and-middle income countries and countries who have been subject to colonialism, it is vital that all participants/collaborators and the

researchers are given equal ownership of the project's outcomes. By working collaboratively alongside a variety of participants, the researchers are able to capture an authentic portrayal of the lives of the resources' end-users, meaning that the resources will be more relevant and useful than if developed by researchers independently. Of course, this is true not only when working in international settings, but also when developing resources for the researchers' home country.

The essence of Co-Creation requires each person to work cooperatively, with the process focusing on the joint effort to create something collaboratively [22]. The 'teamwork' element of the process was frequently reported on in the interviews, with Alex in particular highlighting the importance of 'following people's interests'. The approach of ensuring that what is being Co-Created is of interest to the participants that have been recruited to collaborate, echoes the stances of McNiff and Whitehead and MacDonald who argue that participants are more likely to share knowledge and contribute to the study if they are interested and are involved in the process [38, 39].

'True Co-Creation', in which all stakeholders and end-users play a central-role enables the development of creative and innovative ideas [22]. The collaborators in this study highlighted the importance of working co-operatively and utilising different skill sets of the team in order to induce creativity within the process that otherwise may not have been possible if working on a task independently. Not surprisingly due to the time-poor and heavily target-driven nature of schooling, Emma's approach to Co-Creation involves the teachers taking part in lesson planning and feedback, rather than including students (end-users) in the process. This differs to the other collaborators' approaches and Jansen and Pieters' definition of 'true' Co-Creation, which requires working alongside 'end-users' to develop and improve content. However, on reflection, after working with the Germ's Journey team Emma reported that she is keen to include her students going forward.

The continuous input of diverging groups and individuals, each bringing their own skills sets and having a mutual ownership in the decisions of the process and outcomes is a fundamental element of Co-Creation. Continuous evaluation and inclusive decision making is an important factor within the Co-Creation process, particularly when the outcomes will have a direct impact on the end-user's lives [38, 40]. This conceptualisation is demonstrated in Alex's practice. During the interview, Alex explained that a colleague at the museum had previously explained Co-Creation as a means of evaluating at the beginning and continuing this throughout the process, instead of the more traditional way in which something is created and then is discussed and evaluated at the end of the process.

A central part of the Germ's Journey project included the development of culturally relevant resources for low-and-middle-income countries, including areas of severe poverty. Friere [41] argues that when conducting research concerning 'the oppressed' researchers should work collaboratively alongside this group, so as to gain an insight into and address areas of importance and concern within different communities [42]. Chahel's approach echoes this view, in which they note that by combining the innovative ideas of different people, it enables the "betterment of the community and society". Chahel's charity organisation works within marginalised groups within their local community. By working collaboratively alongside diverse groups, including those in underprivileged communities, the organisation is able to develop beneficial outreach programs and initiatives.

**Empowerment of collaborators in a democratic methodology: Valuing all contributions.** The practice of using Co-Creation with students as partners is reminiscent of Dewey's approach to education, in which he argues that students should play an active role in their learning [23]. This is demonstrated particularly in Ben's practice, in which both staff and students at the university work in partnership in order to create a curriculum and positive student

environment. Synonymously, Ben's process incorporates bell hooks' holistic approach to pedagogy and teaching, in which hooks highlights the benefit of a mutual empowerment between teachers and students [24]. When developing culturally relevant resources for India and Sierra Leone, the Germ's Journey worked in partnership with the end-users (local children, teachers and professionals). It was vital that the collaborators had joint agency and an on-going working relationship with the researchers in order to follow the democratic philosophy underpinning the Co-Creation approach, dismantling traditional researcher-participant relations and instead enabling a shared power in the research process. This is in keeping with the fundamental ethos of theorists such as Friere [41]. It was important for the researchers to be aware that the tensions surrounding Imperialist/Colonial discourse and the White Saviour Complex were deeply rooted within the countries that were part of the Co-Creation process [29, 30]. Areas such as India and Sierra Leone have historically suffered colonialism and have been disempowered, therefore, a constant awareness of these complexities were important when working alongside the collaborators.

## The advantages of Co-Creation

This section explores the advantages of Co-Creation in relation to the literature surrounding the topic and the findings from this study's interviews. The advantages of Co-Creation include the processes' ability to share power within the research process effectively, the positive outcomes of the approach and the effective management of risks and transparency that Co-Creation enables.

**Understanding power relations in research: The use of Co-Creation to mitigate against 'imperialist' discourses.** Being arguably one of its greatest strengths, Co-Creation in its nature is philosophically underpinned by a deeply democratic approach where all collaborators are valued and are an integral part of the process with a shared power and responsibility for the outcomes. In this study the power was distributed across the groups instead of being held centrally by the research team, giving a voice to all collaborators. Their voices were vital, as a way of mitigating against the Imperialist/Colonial discourse and White Saviour Complex [29–32]. Within research, with low to middle-income countries in particular, power is habitually operated in covert ways with the researchers often holding this 'colonialist' power and locus of control. This study wanted to disrupt this historically dominant discourse with the use of the Co-Creation process, enabling for an open and more democratic approach to research.

**Positive outcomes and adding value.** Co-Creation was understood by the collaborators as ensuring better outcomes and provided more meaningful insights. This is supported by Kambil, Ginsberg and Bloch and Kambil, Friesen and Sundaram's original description of Co-Creation, as being a strategy to add value to a business by working alongside their consumers. All collaborators noted the positive outcomes as a result of using Co-Creation within their practices [19, 20].

**Managing risks effectively and enhancing transparency.** The avoidance of risk was another advantage that was reported. Both Dan and Ben commented that the Co-Creation process enables them to decrease the risk of making mistakes during their design process. Furthermore, Dan emphasised the importance of transparency and visibility within the process, (noting the company's use of technology to facilitate this) synonymous of Jansen and Pieters who state that true Co-Creation is a transparent process [22].

## The challenges of Co-Creation

As with any approach or methodology, Co-Creation has its challenges. This section discusses the issues that have arisen for the collaborators in this study, such as timing and organisation

and challenges surrounding contributors and leadership, alongside the most common challenges outlined in existing literature about the approach.

**Timing and organisation issues.**    Literature concerning the challenges of co-creation, highlight that processes such as this 'requires time . . . and sensitivity on the part of the researcher to participants' agendas' (p.40) [39]. This is demonstrated in this study's findings, in which timing and effective organisation were reported to be key factors when working Co-Creatively. Both Alex and Emma commented that the time-intensive nature of Co-Creation can be a major difficulty, especially when working in or with schools, where time is very pressured for teachers and students.

**Leadership demands and ensuring suitable contributors.**    Whilst deeply democratic in approach, co-creation still requires leadership and direction, to avoid the difficulties of managing by committee. In the Co-Creation process, there may be a 'divergence of perspectives, values, and abilities' (p.40) [39] among individuals when collaborating and it may be very challenging to reach an agreement if there are differences of opinions amongst many individuals during the process [38, 43]. Ensuring the right contributors, as well as the risk of involving too many people, is another challenge that the collaborators reported. Alex highlights that it is important to ensure that the 'right people' are chosen when working collaboratively, in that, these people have the correct skill-sets and experiences to be able to assist in a particular task. On a similar note, Dan highlights the risk of involving too many people, commenting that if everybody was involved in everything that was being developed, it would be impossible to complete a task. Likewise, Emma emphasises the challenge of finding a balance between involving the contributions of many people and needing strong leadership to guide the process.

Furthermore, Ben commented that one of the trials that his practice has faced, is the commitment of partners/stakeholders. Gillis and Jackson also recognise this as a challenge, highlighting that by including many people, it may be difficult for a project to maintain the commitment from them over time. Local social and structural aspects also need consideration early in projects [43]. For example, hand-hygiene interventions cannot succeed if fundamental infrastructure is not in place to support the practices which are being promoted.

## Conclusion

By considering the differing definitions and approaches, and the advantages and challenges of Co-Creation as discussed by the five collaborators involved in the Germ's Journey project and the literature surrounding the topic, a clearer evaluation can be made.

Firstly, by exploring how each collaborator uses Co-Creation within their practice, recommendations can be made as to how to successfully carry-out the process; from suggestions about ensuring the suitability of the co-creators to specific software programmes to aid the process.

It is important to note that although the collaborators involved in the interviews in this study all held senior positions, data was also collected through focus groups (N = 37) with the teachers in Sierra Lone and questionnaires (N = 129) with the teachers in India and the UK who, among children are also the 'end-users' of the resources. It was decided by the researchers that the children, although involved in the development of the original UK resources, would not be interviewed regarding their opinions of Co-Creation. The target age of the children who the resources are designed for are very young and it was decided that this topic would be too complex to understand and discuss.

The Co-Creation process undertaken by the Germ's Journey research team when developing the educational resources involved the continuous collaboration and contributions of

parents and children. For example, when developing the original UK book, research was conducted with parents and children in order to gather opinions on the types of illustrations and colours that the book should contain. This feedback also shaped the format and storyline of book. Once published, a study [1] was also conducted to gather parents and teachers' feedback on the UK book. Following this research, and having already established alongside children and parents the format and illustrations of the UK book, when developing resources for India and Sierra Leone, the resources were co-created alongside the collaborators and teachers to ensure they were culturally relevant.

Although parents were involved in the initial design of the UK resources, they were not used as collaborators throughout the whole development process. This was mainly due to limited access to parents within India and Sierra Leone. However, as a study that investigated the impact on parental co-creation within primary educational innovations in India has indicated: 'parental involvement in the execution stage of the initiatives impacts their perceived value more than at the conceptualization stage' [44]. Arguably, the impact of parents' involvement in co-creation has more value during the implementation stage rather than the development stage of an intervention. The researchers' focus for this study was to implement the resources into schools and community centres. However, should the resources be implemented in domestic and home environments in the future studies, parental involvement would then be required.

Having worked in Sierra Leone and India, the researchers acknowledge the strong 'social norm culture', further recommendations for future work include developing measures to ensure accountability to populations and work towards a more robust community action plan model. Previous epidemics of Ebola in Sierra Leone and Polio in India have had an impact on the community and their behaviours, and their views towards aid workers during the epidemics/pandemics; it was important for the researchers to guard against issues of the White Saviour Complex and Imperialist/Colonial discord, hence the use of Co-Creation. Within the time-frame of this research such robust community feedback loop measures were not possible, but for future work these issues will be addressed to ensure greater engagement with the wider community to ensure sustainability of the project.

The resources have led to increased understanding [1, 2], change in behaviour [45] and reduction in illness.

With regards to the Germ's Journey project, without the contributions of each of the collaborators, their teams, and children and teachers, the resources would have been developed using only the knowledge and abilities of the research team. Each of the resources would have been created with limited information and potential bias. For example, when creating resources for India and Sierra Leone it would have been ill-judged to attempt to develop resources in areas where the researchers do not live, risking tensions surrounding Imperialist/Colonial discourse and the White Saviour Complex. Therefore, it was integral to the Germ's Journey project, to utilise the methodology of Co-Creation, to develop the educational resources and to evaluate their effectiveness in engaging and aiding understanding in the topic of germs and hand-hygiene. Furthermore, by adopting a Co-Creation approach, the on-going relationships with each collaborator ensures the shared-ownership and sustainability of the resources and enables the opportunity for further resources to be continually adapted and developed for their contexts.

Although, as noted by the five collaborators in this study and in the literature surrounding this methodology, Co-Creation is not without its challenges (such as the time and organisation intensive nature of the process), this does not outweigh its ability to produce an effective, -quality end result. By Co-Creating resources with collaborators in-situ, the resources will be more culturally relevant and therefore more likely to be used and engaged with. This is crucial

during a global pandemic where, as WHO state, handwashing is still the most effective measure to manage the transmission of COVID-19 [46].

By discussing the approach with collaborators from a diverse range of contexts, (including a museum, educational institutions, a commercial business company and a charity organisation) across three countries, an understanding and evaluation can be made of the nuanced approaches and end-results of Co-Creation, as well as the shared advantages and challenges recognised by all five collaborators.

## Acknowledgments

The collaborators, teachers, parents and children at the participating schools in the Midlands, England, Thinktank Birmingham Science Museum, University of Makeni, Manav Sadhna and PAL International are thanked for their on-going involvement and commitment to the project.

## Author Contributions

**Conceptualization:** Sapphire Crosby, Sarah Younie, Katie Laird.

**Data curation:** Sapphire Crosby, Sarah Younie, Katie Laird.

**Formal analysis:** Sapphire Crosby, Sarah Younie, Iain Williamson, Katie Laird.

**Investigation:** Sapphire Crosby, Sarah Younie, Katie Laird.

**Methodology:** Sapphire Crosby, Sarah Younie, Iain Williamson, Katie Laird.

**Project administration:** Sapphire Crosby, Sarah Younie, Katie Laird.

**Resources:** Sapphire Crosby, Sarah Younie, Iain Williamson, Katie Laird.

**Supervision:** Sarah Younie, Iain Williamson, Katie Laird.

**Writing – original draft:** Sapphire Crosby.

**Writing – review & editing:** Sapphire Crosby, Sarah Younie, Iain Williamson, Katie Laird.

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
