## [Decision Letter · Decision Letter 0]

29 Jul 2020

PONE-D-20-13329

Evaluating approaches to designing effective Co-Created hand-hygiene interventions for children in India, Sierra Leone and the UK.

PLOS ONE

Dear Dr. Sapphire Crosby,

Thank you for submitting your manuscript to PLOS ONE. After careful consideration, we feel that it has merit but does not fully meet PLOS ONE’s publication criteria as it currently stands. Therefore, we invite you to submit a revised version of the manuscript that addresses the points raised during the review process.

We look forward to receiving your revised manuscript.

Kind regards,

Francesco Di Gennaro

Academic Editor

PLOS ONE

Journal Requirements:

Additional Editor Comments (if provided):

Dear authors.

Congratulations.

I find your manuscript very important, and very well wrote.

I suggest also to introduce "The concept of “children at risk” , as reported in others studies ( doi:10.3390/ijerph15071350 and doi: 10.1017/S2040174415001427.) that underlines how this vulnerable group needs medical but also scientific attention.

Reviewers' comments:

Reviewer's Responses to Questions

**Comments to the Author**

1. Is the manuscript technically sound, and do the data support the conclusions?

Reviewer #1: Yes

Reviewer #2: Yes

2. Has the statistical analysis been performed appropriately and rigorously? 

Reviewer #1: Yes

Reviewer #2: Yes

3. Have the authors made all data underlying the findings in their manuscript fully available?

Reviewer #1: Yes

Reviewer #2: Yes

4. Is the manuscript presented in an intelligible fashion and written in standard English?

Reviewer #1: Yes

Reviewer #2: Yes

5. Review Comments to the Author

Reviewer #1: I have read through the manuscript. The research was rigorously conducted and it cut across three continents.

1. The concept of co-creation is relatively new, and the manuscript answers the question of acceptability and adaptability of this concept in Low and middle income countries. Power sharing is a big deal in social marketing of behavioral change; if an equilibrium is not reach, no matter how beautiful an idea or a project is, it will end up as an effort in futility. The manuscript by the researchers was scientifically rigorous; bringing forth participants from three continents with each bringing forth his or her concept of co-creation, the advantages and the problems associated with it.

2. The research was a qualitative work and the analysis of the findings was purely thematic. Unfortunately, there were very few journals on the concept of co-creation that were reviewed. The research question did not seek to establish any statistical significance between the experiences of the participants but rather to simply lay-out the experiences of the co-creation partners on children hand washing intervention in their various countries. The process of thematic analysis of the response of the participants was rigorous enough to answer the research question.

3. The authors provided enough data underlying the findings available.

4. The manuscript was provided in a simple, understandable and intelligible fashion and it was written in a standard English.

My additional comments/clarifications are:

1. Power sharing (line 749-754): The author failed to document the views and opinions of the children about the hand washing intervention in the Low and Middle income countries. The co-creation tend to revolve around the teachers with very little said about the children who will be the end users of the intervention.

2. In the creation of "Germ's Journey Book" in England, the children were part of the creation and that made the acceptance very easy. What was the children view about co-creation in India and Sierra Leone? What was their perceived advantages and pit-falls of co-creation? The manuscript failed to show if they were interviewed or not. This should be included in the final draft. Perhaps, the research was not designed to hear their views. However, as the supposed end-users of the intervention, their view should be integrated into the study.

3. I was surprised parents were not considered as collaborators in co-creation. Their views and opinion will go a long way in consolidating the hand washing intervention at home. As rigorous and tasking as it is in organizing co-creation, I strongly believe co-creation should be a "web concept"as against the "near-linear approach" in the study. The opinion of the selected children and parents would have added more credence to the study.

That being said, the manuscript was very informative, scientific and I found it a good read.

Reviewer #2: In my opinion, this is a good manuscript. India and SL both countries where I have worked in, have a strong social norm culture that impacts practically everything an individual does. The manuscript would have benefitted more in terms of ‘Accountability to populations’ and what are some of the measures in place to see if the ‘community feedback loop is closed and community voices heard’ especially around Education and WASH and the filling the gap between the demand and supply side of services. Community Engagement, meaningful participation or community action planning – would have been some way forward to ensure sustainability at all levels. What I would also like to point out is that both countries have been targets to some of the Global Pandemics like Ebola and Polio and that also has an impact on the community and it’s behaviours. Overall, the manuscript is well written and highly appreciated.

6. PLOS authors have the option to publish the peer review history of their article (what does this mean?). If published, this will include your full peer review and any attached files.

Reviewer #1: **Yes: **Dr. Atilola A. Adeleke

Reviewer #2: **Yes: **Aarunima Bhatnagar

---

## [Author Response · Author response to Decision Letter 0]

19 Aug 2020

The authors would like to thank the reviewers for their helpful feedback of the manuscript (“Evaluating approaches to designing effective co-created hand-hygiene interventions for children in India, Sierra Leone and the UK.”), and have addressed the recommended amendments accordingly. 

As requested, two copies of the manuscript have been uploaded, one with tracked changes and a clean copy without. 

Please find below the reviewers’ comments and the authors’ responses:

Reviewer #1: 

My additional comments/clarifications are:

1. Power sharing (line 749-754): The author failed to document the views and opinions of the children about the hand washing intervention in the Low and Middle income countries. The co-creation tend to revolve around the teachers with very little said about the children who will be the end users of the intervention. 

2. In the creation of "Germ's Journey Book" in England, the children were part of the creation and that made the acceptance very easy. What was the children view about co-creation in India and Sierra Leone? What was their perceived advantages and pit-falls of co-creation? The manuscript failed to show if they were interviewed or not. This should be included in the final draft. Perhaps, the research was not designed to hear their views. However, as the supposed end-users of the intervention, their view should be integrated into the study. 

-> The following text has been added for further clarity for points 1 and 2: 

“It was decided by the researchers that the children, although involved in the development of the original UK resources, would not be interviewed regarding their opinions of Co-Creation. The target age of the children who the resources are designed for are very young and it was decided that this topic would be too complex to understand and discuss.” (Line 752-755).

3. I was surprised parents were not considered as collaborators in co-creation. Their views and opinion will go a long way in consolidating the hand washing intervention at home. As rigorous and tasking as it is in organizing co-creation, I strongly believe co-creation should be a "web concept"as against the "near-linear approach" in the study. The opinion of the selected children and parents would have added more credence to the study.That being said, the manuscript was very informative, scientific and I found it a good read. 

-> The following text has been added for further clarity:

“For example, when developing the original UK book, research was conducted with parents and children in order to gather opinions on the types of illustrations and colours that the book should contain. This feedback also shaped the format and storyline of book. Once published, a study [1] was also conducted to gather parents and teachers’ feedback on the UK book. Following this research, and having already established alongside children and parents the format and illustrations of the UK book, when developing resources for India and Sierra Leone, the resources were co-created alongside the collaborators and teachers to ensure they were culturally relevant. 

Although parents were involved in the initial design of the UK resources, they were not used as collaborators throughout the whole development process. This was mainly due to limited access to parents within India and Sierra Leone. However, as a study that investigated the impact on parental co-creation within primary educational innovations in India has indicated: ‘parental involvement in the execution stage of the initiatives impacts their perceived value more than at the conceptualization stage’ [46]. Arguably, the impact of parents’ involvement in co-creation has more value during the implementation stage rather than the development stage of an intervention. The researchers’ focus for this study was to implement the resources into schools and community centres. However, should the resources be implemented in domestic and home environments in the future studies, parental involvement would then be required.” (Line 758-775). 

Reviewer #2: In my opinion, this is a good manuscript. India and SL both countries where I have worked in, have a strong social norm culture that impacts practically everything an individual does. The manuscript would have benefitted more in terms of ‘Accountability to populations’ and what are some of the measures in place to see if the ‘community feedback loop is closed and community voices heard’ especially around Education and WASH and the filling the gap between the demand and supply side of services. Community Engagement, meaningful participation or community action planning – would have been some way forward to ensure sustainability at all levels. What I would also like to point out is that both countries have been targets to some of the Global Pandemics like Ebola and Polio and that also has an impact on the community and it’s behaviours. Overall, the manuscript is well written and highly appreciated. 

-> The following text has been added for further clarity:

“Having worked in Sierra Leone and India, the researchers acknowledge the strong ‘social norm culture’, further recommendations for future work include developing measures to ensure accountability to populations and work towards a more robust community action plan model. Previous epidemics of Ebola in Sierra Leone and Polio in India have had an impact on the community and their behaviours, and their views towards aid workers during the epidemics/pandemics; it was important for the researchers to guard against issues of the White Saviour Complex and Imperialist/Colonial discord, hence the use of Co-Creation. Within the time-frame of this research such robust community feedback loop measures were not possible, but for future work these issues will be addressed to ensure greater engagement with the wider community to ensure sustainability of the project.” (Line 776 – 785). 

I hope that these amendments are sufficient in order for the article to be published in due course.

Yours Faithfully,

Sapphire Crosby (corresponding author) 

p13212007@my365.dmu.ac.uk

---

## [Decision Letter · Decision Letter 1]

2 Sep 2020

Evaluating approaches to designing effective Co-Created hand-hygiene interventions for children in India, Sierra Leone and the UK.

PONE-D-20-13329R1

Dear Dr.ssa Crosby,

We’re pleased to inform you that your manuscript has been judged scientifically suitable for publication and will be formally accepted for publication once it meets all outstanding technical requirements.

Kind regards,

Francesco Di Gennaro

Academic Editor

PLOS ONE

Additional Editor Comments (optional):

Dear Authors,

congratualtion for your manuscript!

Reviewers' comments:

Reviewer's Responses to Questions

**Comments to the Author**

1. If the authors have adequately addressed your comments raised in a previous round of review and you feel that this manuscript is now acceptable for publication, you may indicate that here to bypass the “Comments to the Author” section, enter your conflict of interest statement in the “Confidential to Editor” section, and submit your "Accept" recommendation.

Reviewer #1: All comments have been addressed

Reviewer #2: All comments have been addressed

2. Is the manuscript technically sound, and do the data support the conclusions?

Reviewer #1: Yes

Reviewer #2: Yes

3. Has the statistical analysis been performed appropriately and rigorously? 

Reviewer #1: Yes

Reviewer #2: Yes

4. Have the authors made all data underlying the findings in their manuscript fully available?

Reviewer #1: Yes

Reviewer #2: Yes

5. Is the manuscript presented in an intelligible fashion and written in standard English?

Reviewer #1: Yes

Reviewer #2: Yes

6. Review Comments to the Author

Reviewer #1: The researchers have done an excellent job. All the initial concerns and recommendations has been addressed. Congratulations to the team.

Reviewer #2: It's a well written and sound research paper with due diligence. The data has been presented in a user friendly format.

7. PLOS authors have the option to publish the peer review history of their article (what does this mean?). If published, this will include your full peer review and any attached files.

Reviewer #1: **Yes: **Dr. Atilola Atilade Adeleke.

Reviewer #2: **Yes: **Aarunima Bhatnagar

---

## [Editor Report · Acceptance letter]

7 Sep 2020

PONE-D-20-13329R1 

Evaluating approaches to designing effective Co-Created hand-hygiene interventions for children in India, Sierra Leone and the UK. 

Dear Dr. Crosby:

I'm pleased to inform you that your manuscript has been deemed suitable for publication in PLOS ONE. Congratulations! Your manuscript is now with our production department. 

Kind regards, 

on behalf of

Dr. Francesco Di Gennaro 

Academic Editor

PLOS ONE